# Local Agricultural Management Filters Morphological Traits of the South American Palm Weevil (*Rhynchophorus palmarum* L.; Coleoptera: Curculionidae) in Ornamental Palm Plantations

Moises Ponce-Méndez [1], Miguel A. García-Martínez [1,*][ID], Ricardo Serna-Lagunes [1][ID], Rodrigo Lasa-Covarrubias [2][ID], Ehdibaldo Presa-Parra [1][ID], Joaquin Murguía-González [1][ID] and Carlos Llarena-Hernández [1]

[1] Facultad de Ciencias Biológicas y Agropecuarias Región Orizaba-Córdoba, Universidad Veracruzana, Josefa Ortiz de Domínguez S/N, Peñuela, Amatlan de los Reyes 94945, Veracruz, Mexico
[2] Red de Manejo Biorracional de Plagas y Vectores, Instituto de Ecologia, A.C., Xalapa 91073, Veracruz, Mexico
[*] Correspondence: miguelgarcia05@uv.mx

**Abstract:** Insect pests show phenotypic plasticity as a function of resource availability and limiting conditions. Although *Rhynchophorus palmarum* displays high variation in certain morphological traits, it is still not clear how and which of these are being filtered along agricultural management gradients in palm plantations. This study assesses the influence of biophysical structure of ornamental palm plantations and agrochemical use on morphological traits of adults in 15 permanent plots of ornamental palm plantations in Veracruz, Mexico. A total of 4972 adults were and their body length, pronotum width, rostrum length, and mesothorax depth were measured. Body length and mesothorax depth of adults of both sexes were greater in plantations with a high diversity of palm species and frequency of fertilizer use. Rostrum length of females increased as a function of palm density, and pronotum width of both sexes was positively related with the use of insecticides. Local characteristics of agricultural management of palm plantations might filter integrated, adaptative, and environment-specific phenotypes. This is the first ecological study of the south American palm weevil that provides new insights on the current intensive management of ornamental palm plantations that far from controlling, benefits current geographic expansion, demographic outbreak, and economic impact of this pest.

**Keywords:** morphometric measurements; phenotypic variation; phenotypic plasticity; agrochemical use; crop structure; black weevil; phytophagous insect; crop diversity; local driver; pest ecology

## 1. Introduction

Agriculture is one of the most important production sectors worldwide, and particularly its management intensity at the local field scale plays a key role for maximizing yield and profitability of crop production [1]. Locally, the most significant descriptors of management intensity are the biophysical structure of plantations (i.e., density and diversity of crops), and the frequency of application of synthetic agrochemicals (i.e., insecticides, herbicides, fungicides, and fertilizers) [2]. The high-intensity management of plantations are characterized by a simplified structure and a frequent, intense agrochemical use due to the high demand for crop productivity by the growing world human population [3,4]. Despite agricultural intensification being an effective management strategy at the initial stage, currently it has adversely affected the environment and human health. As a result, its efficacy in pest control has decreased [5–7].

Intensive agriculture as it is currently practiced has proven to increase the local abundance of wide-tolerant herbivores into pest-level populations [8]. This is due to the monoculture planting, which includes the removal of plant species used as cover cropping, hedgerows, border plantings, and grass strips, which enables native and invasive insect

pests to spread to new distribution areas [9]. Currently, greater amounts of insecticides are persistently applied and become a key component of pest management programs to avoid pest population outbreaks [10,11]. Nevertheless, widespread use of broad-spectrum, very toxic chemical insecticides with low specificity may lead to long-term environmental and phytosanitary risks, potentially resulting in the development of populations of resistant insects [12–14]. For this reason, high-intensity management might act as an artificial selection pressure, since pests have filtered their population attributes, spatial and temporal distribution, and even certain morphological traits in several plantations [15,16].

Pest insect populations show phenotypic plasticity (i.e., the ability of individual genotypes to produce different morphological characteristics in different environments) as a function of resource availability and limiting conditions of the local habitat [17]. These changes are mainly shaped by host and diet preferences and intra- or interspecific competition [18–21]. Due to this adaptability, several populations become abundant pests in highly managed plantations [14]. Some studies have demonstrated that intraspecific variation in morphological traits of insect pests generally indicate functional traits such as reproductive, feeding and displacement [22–24]. Thus, changes in morphological traits may provide adaptability to insect pest populations, which increase damage and economic impact in important crops worldwide [25].

The south American palm weevil, *Rhynchophorus palmarum* (L.) (Coleoptera: Curculionidae), is an invasive and polyphagous pest in the Americas [26–30]. Adults have been reported as phytophagous of economically important crops such as avocado (*Persea americana* Mill.; Laurales: Lauraceae), banana (*Musa* × *paradisiaca* L.; Zingiberales: Musaceae), cocoa (*Theobroma cacao* L.; Malvaceae: Theobromeae) and sugar cane (*Saccharum officinarum* L.; Poales: Poaceae) [31,32]. However, it has been highlighted as the primary pest of palms (Arecales: Arecaceae) cultivated for food or ornamental purposes [32]. The larval phase of this weevil directly damages palms because they feed on meristematic tissue and can tunnel palm stems. This prevents palms from producing new fronds, destroys the growth point, damages the lower stem and rhizomes, and eventually causes palm death [28,33]. Adults also indirectly damage them since they act as vectors of the causal agents of the bud rot disease (*Phytophthora palmivora* Butler; Peronosporales: Pythiaceae), and the red ring disease (*Bursaphelenchus cocophilus* (Cobb); Tylenchida: Aphelenchoididae) [29,30,34]. Given this biological, ecological, and phytosanitary importance, *R. palmarum* represents a devastation pest in several palm plantations, mainly ornamental ones.

The commercial exploitation of ornamental palms plays a valuable role in the economy of international trade [35]. Plantations represent a significant source of income for small farmers due to palm health, quality, and high aesthetic value [36]. This crop is rare throughout Mexico as more producers are entering the market. Their current planted area is only represented in Baja California Sur (82 ha), Chiapas (16 ha), San Luis Potosí (734 ha), and Veracruz (711 ha) with ca. 468 cultivated palm species. Particularly, in Veracruz, there are approximately 40 cultivated palm species distributed in 19 municipalities, where there are small farms ranging from 0.56 to 3.8 ha. The total area of this crop is 1559 ha and belongs to 2768 farms. The ornamental palms are commercialized to coastal touristic areas in Mexico [36]. However, ornamental palm agribusinesses, and particularly those of central Veracruz, have reported yield losses of approximately $6000.00 USD/ha caused by *R. palmarum* [37]. Although the management of this pest has intensified the use of broad-spectrum insecticides mixed with synthetic baits and/or food attractants [37–39], populations are currently expanding their geographical range and exploiting novel palm hosts [40–42]. In fact, the density of certain ornamental palm species may modulate the abundance population of this pest in space and time [41]. Given this context, agrochemical use and crop density and diversity in ornamental palm plantations might improve *R. palmarum* populations, particularly their morphological traits [43], which promotes their survival, spread and establishment in vast agricultural areas.

The variation in morphological traits of *R. palmarum* individuals is poorly understood since there are only a few preliminary studies [43,44]. Although individuals display high

phenotypic plasticity in certain morphological traits [43,44], it is still not clear how and which of them are being filtered along agricultural management gradients in palm plantations. This may happen as a result of resource availability and experienced conditions, causing a constraint in individuals' tolerance and conferring environmental determination or polyphenism to the populations of this pest [45]. Therefore, the influence of agrochemical use, and crop density and diversity that could potentially filter morphological traits in ornamental palm plantations, could be an important aspect to consider in agricultural management planning in order to have a broader perspective of the success of *R. palmarum* populations.

Elucidating the effect of local agricultural management on the variation in morphological traits of *R. palmarum* populations may help to explain the ongoing spread and the ineffectiveness of different controlling strategies of this pest [32,46]. This study assesses the influence of local agricultural management, measured as a biophysical structure of ornamental palm plantations, and agrochemical use on morphological traits of adult populations. First, we analyzed the variation in five morphometric measurements of this pest populations, and then we assessed if local characteristics, reflecting different levels of intensity of agricultural management, influenced the variation in these morphological traits. We postulated that the density and the diversity of palms and the use of insecticides and fertilizers may have disproportionate benefits for different morphological phenotypes in *R. palmarum* populations.

## 2. Materials and Methods

### 2.1. Study Area

Field work was conducted in the municipalities of Córdoba, Fortín, Ixtaczoquitlán, and Naranjal on the central plain of the great mountain region in Veracruz, Mexico, in the mid-watershed of the Jamapa river (Figure 1). The climate is warm and humid throughout the year, with a mean total annual precipitation of 2199 mm and a mean annual temperature of 20 °C. The dry, warm season occurs from March to June (mean monthly precipitation = 85 mm), the rainy, warm season occurs from July to October (mean monthly precipitation = 201 mm), and the relatively dry, cool season occurs from November to February (mean monthly precipitation = 26 mm) [47]. Originally the most common vegetation type in the area was tropical dry forest, or tropical semi-deciduous forest [48]. However, the region is a mosaic of land uses/covers such as secondary forests, sugarcane, vegetable and ornamental crops, and human settlements and infrastructure [49].

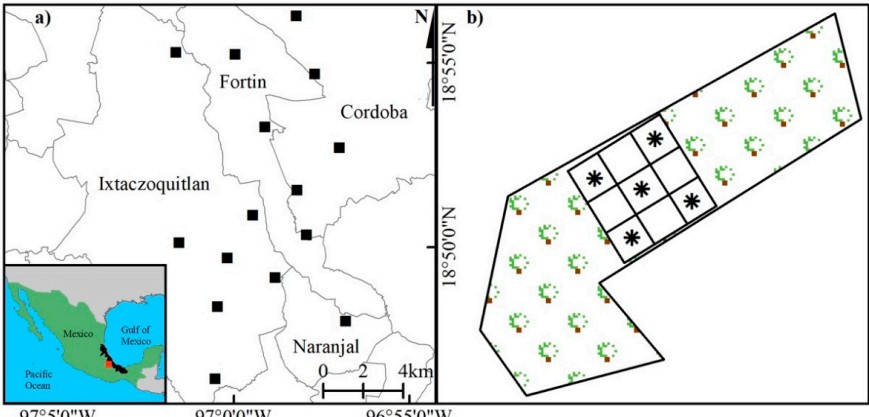

**Figure 1.** Location of the study area in central Veracruz, Mexico. (**a**) The location of the study area (red square) is indicated in the inset map along with the state of Veracruz (black polygon), and the location of Mexico (green polygon) between North and Central America (gray polygons). The black squares represent the 15 permanent sampling plots, and political division is shown as a solid gray line. (**b**) Sampling design: where a permanent sampling plot is represented with the five selected sampling units in an ornamental palm plantation.

## 2.2. Sampling Plot Selection and Design

In the study area, we selected a total of 15 permanent sampling plots of $100 \times 100$ m, embedded in ornamental palm plantations with an area ranging from 1.3 to 3.7 ha (Figure 1b). Palm plantations have been used for cultivating a different composition of ornamental palms species for more than 10 years (Table S1). These represented a gradient of agricultural management intensity. Patterns of land use and land cover in the study area, the accessibility granted by the owners and limited access to several zones precluded a balanced design and equidistant sampling plot selection. Selected sampling plots were separated by a distance ranging from 2 to 15 km, between 793 and 803 m a.s.l. Each 1 ha-permanent sampling plot was divided into nine quadrats ($\approx 33.33 \times 33.33$ m), and only five of them, distributed systematically in the center and in the four corners of the plot (marked with asterisks in Figure 1b), were used as sampling units.

## 2.3. Local Agricultural Management Characterization

Agricultural management in each sampling plot was estimated considering the biophysical structure of plantation and agrochemical used to maintain the crop productivity. We estimated palm density (palms/ha), palm diversity (palm species/ha), and the abundance of the queen palm (*Syagrus romanzoffiana* (Cham.) Glassman; Arecales: Arecaceae), and the pygmy date palm (*Phoenix roebelenii* O'Brien; Arecales: Arecaceae). Both palm species have been previously reported as hosts of *R. palmarum* [41], and they are the dominant palm species in the study area (Table S1). Then, a semi-structured interview with farm managers and/or owners was conducted to identify the frequency of application of insecticides (1-L bottles), herbicides (1-L bottles), fungicides (1-L bottles), and fertilizers (10-kg bags) per year (Table S2). We did not collect data on the usage frequency or quantity of certain commercial brands of chemicals

## 2.4. Population Adult Survey

A total of five traps, one per each selected sampling unit, were set in each sampling plot. For sampling adult populations, a novel and affordable food-attractant trap combination was set (Figure 2), which had previously been highly effective in the study area [37]. These traps were constructed using 3 L colorless polyethylene bottles (PepsiCo® Inc., Mexico City, Mexico). For adult weevil access, two square holes of $4 \times 4$ cm were cut into the sides of the bottle at a height of 20 cm from the base. Only three sides of the square opening were cut, as the upper side of the square was left uncut to avoid entry of excess rainwater. These trap openings simulated a ramp of 45° perpendicular to the axis of the bottle. In addition, five 6-mm-diameter holes were cut around the perimeter of the bottle at a height of 5 cm from the base to prevent the potential accumulation of rainwater. The food-attractant was a mixture of banana (*Musa paradisiaca* L.; Zingiberales: Musaceae), pineapple (*Ananas comosus* Merr.; Poales: Bromeliaceae), sugarcane (*Saccharum officinarum* L.; Poales: Poaceae) and sugarcane molasses [39]. The food attractant mixture was placed inside each trap (200 g of mixture and 200 mL of water). These amounts were enough to guarantee that the food-attractant would ferment and attract weevils for a month [37,39]. The food-attractant was replaced with a fresh attractant every 4 weeks. Each trap was set approximately in the center of the sampling unit and hung from a palm frond using a rope tied to the neck of the bottle [50]. To ensure that weevils would enter the traps after perching on palm stems, the hanged traps were carefully adjusted to be in direct contact with the stem of the selected palm at a height between 1.5 and 2 m above ground level. Traps were set at the beginning of every month and revised after 4 weeks of exposure in the field, and captured adult weevils were placed in 70% alcohol and taken to the laboratory. The monitoring survey lasted a whole year from June 2020 to July 2021. In each plot 5 traps were set; considering there were 15 sampling plots, a total of 75 traps were set each month, resulting in a total of 900 traps for the entire sampling of 12 months.

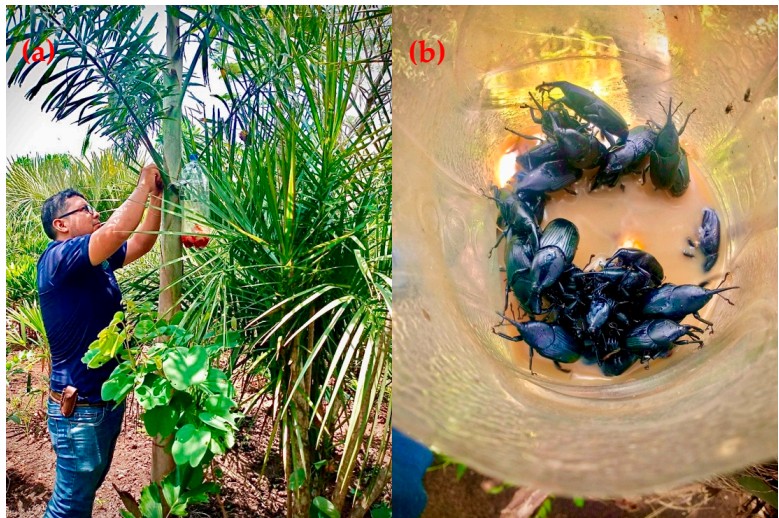

**Figure 2.** Frontal view of the set traps in 15 ornamental palm plantations in central Veracruz, Mexico. (**a**) The first author is hanging a trap from a palm. (**b**) Collected adult weevils after 4 weeks of trap-food-attractant exposure in an ornamental palm plantation.

### 2.5. Morphometric Trait Measurements

All captured weevils, in each sampling plot per month, were counted and sorted by sex. Several morphometric traits were measured for each captured weevil: (1) body length (distance from the anterior end of the pronotum to the posterior end of the pygidium in dorsal view), which indicates reproductive advantage such as fecundity and mating capacity [51,52] (Figure 3a); (2) pronotum width (distance at the widest point on the pronotum in dorsal view) [53], which has been suggested as a lifespan population predictor [54] (Figure 3b); (3) elytron length (distance between base and apex in dorsal view), that reflects fly capacity [51] (Figure 3c); (4) rostrum length (distance from the ocular sulcus to the apex of the rostrum or the tip of epistoma) suggested as an indicator of adaptation to the oviposition site [55,56] (Figure 3d); (5) mesothorax depth (distance between the lower and the upper surface at the base level of elytra in profile view), which has been related to the weight and quality of development of weevils [56] (Figure 3e). Morphometric measurements were obtained from digital images obtained with an optical microscope (Leica Z16 APOA, DMC 2900, LS Score), using the software Leica LAS EZ, and analyzed in ImageJ software version 1.46r. Before they were photographed, each weevil was gripped at the body with forceps and fixed on the object stage.

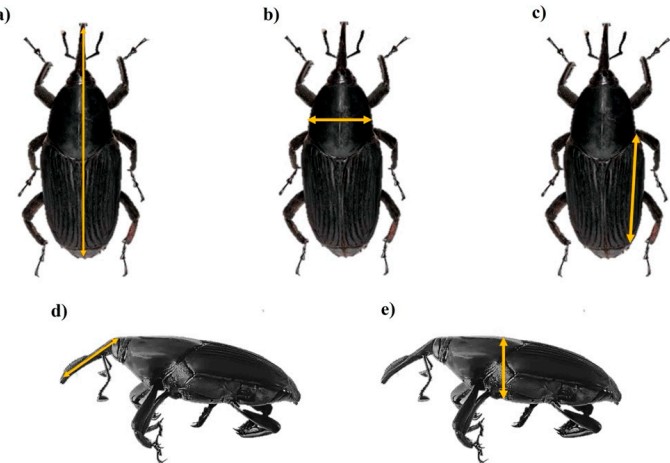

**Figure 3.** Images of the *Rhynchophorus palmarum* male showing morphometric trait measurements taken for (**a**) body length, (**b**) pronotum width, (**c**) elytron length, (**d**) rostrum length, and (**e**) mesothorax depth.

*2.6. Data Analysis*

Due to sexual dimorphism on the *R. palmarum* species [37,41,46], we tested whether morphological traits varied across sexes and plots with general linear mixed models. We specified plot, sex, and the two-way interactions as fixed effects, and the total adults captured per month as a random factor. Model selection was based on *P* values, starting with the full model, and dropping terms for which model comparison using chi-squared tests gave $p > 0.05$ [24]. A Gaussian error distribution was appropriate for all response variables after testing for normality (Shapiro–Wilk test).

In order to identify the influence of local management characteristics in the morphological traits of *R. palmarum* populations, we followed univariate selection procedures for regression-based models. As these statistical techniques are sensitive to collinearity between predictor variables (local management characteristics), we used the Spearman correlation coefficient to exclude correlated variables [57]. For each set of significantly correlated variables, we retained only one that was considered to be the most intuitive and interpretable.

Generalized linear models were used to assess the independent effects of each local management characteristic on each morphological trait (i.e., a single univariate regression between a response and a predictor variable). We applied a Gaussian error distribution for all response variables (i.e., morphological traits) after testing for normality (Shapiro–Wilk test). The goodness-of-fit of the models was estimated as the explained deviance for each model using the *modEvA* package [58]. All analyses were run using the R software (version 4.1.3) [59], and the packages *nlme* [60], and *lme4* [61].

**3. Results**

*3.1. Variation in the Morphological Traits*

A total of 4972 *R. palmarum* adults were captured across the 15 ornamental palm plantations during a whole year of sampling. A total of 3077 (61.9%) of the captured weevils were male, and 1895 (38.1%) were female. The proportion of traps that did not capture adults was less than 9% of all set traps during the entire sampling period ($n = 900$ traps). Considering all of the captured weevils, body length ($F_{1,4958} = 11.6$, $p < 0.001$) varied due to interaction between the factors of plot and sex. It varied significantly from 34.27 to 39.35 mm among plots and was significantly greater in males than females (Table 1). Pronotum width varied by the independent effects of the factors of plot ($F_{1,4958} = 11.1$, $p < 0.001$), and sex ($F_{1,4958} = 20.8$, $p < 0.001$). This trait varied significantly from 7.71 to 14.48 mm among plots and was significantly greater in males than females. Elytron length only varied between sexes ($F_{1,4958} = 21.8$, $p < 0.001$), and this measurement was greater in males than in females. Rostrum length ($F_{1,4958} = 46.2$, $p < 0.001$), and mesothorax depth ($F_{1,4958} = 37.1$, $p < 0.001$) were affected by the interaction between the factors of plot and sex. Rostrum length varied from 11.69 to 13.98 mm, meanwhile mesothorax depth from 5.31 to 6.43 mm. Both traits were significantly greater in males than in females.

**Table 1.** Variation in morphological traits (mean ± SD) of *R. palmarum* adult populations captured in ornamental palm plantations with different intensity of agricultural management in central Veracruz, Mexico.

| Morphometric Measuments (mm) | Males | Females | Both Sexes |
|:---:|:---:|:---:|:---:|
| Body length | 38.32 ± 0.61 | 36.09 ± 1.13 | 37.21 ± 1.44 |
| Pronotum width | 12.45 ± 1.44 | 9.61 ± 1.81 | 11.02 ± 2.16 |
| Elytron length | 20.95 ± 0.16 | 20.27 ± 0.49 | 20.61 ± 0.49 |
| Rostrum length | 13.31 ± 0.31 | 12.43 ± 0.53 | 12.87 ± 0.61 |
| Mesothorax depth | 6.11 ± 0.14 | 5.69 ± 0.31 | 5.9 ± 0.31 |
| Abundance (individuals) | 3077 | 1895 | 4972 |

### 3.2. Local Management Characteristics

In the ornamental palm plantations studied, palm density varied from 943 to 2331 palms/ha (Table 2), and it was positively correlated with the abundance of the queen palm (*S. romanzoffiana*) ($\rho = 0.76$, $p = 0.02$). Palm diversity varied from 1 to 8 palm species/ha, and the abundance of the pygmy date palm (*P. roebelenii*) from 0 to 1242 individuals. They were not significantly correlated between them or to any characteristic ($p > 0.05$). Regarding agrochemical use, the use of insecticides varied from 0 to 88 1-L-bottles/year, and it was correlated with the use of herbicides ($\rho = 0.92$, $p < 0.001$) and fungicides ($\rho = 0.88$, $p < 0.001$). The use of fertilizers varied from 0 to 104 10-kg-bags/year, and it was not significantly correlated to any characteristic.

**Table 2.** Local characteristics of agricultural management in ornamental palm plantations in central Veracruz, Mexico.

| Management Descriptor | Mean ± SE | Range | Mode |
|---|---|---|---|
| (a) Biophysical structure | | | |
| Palm density (palms/ha) | 1508.87 ± 115.77 | 943–2331 | 1260 |
| Palm diversity (palm species/ha) | 4.93 ± 0.51 | 1–8 | 6 |
| *S. romanzoffiana* abundance (individuals) | 724.67 ± 99.85 | 0–1602 | 0 |
| *P. roebelenii* abundance (individuals) | 295.20 ± 77.75 | 0–1242 | 135 |
| (b) Agrochemical use | | | |
| Insecticides (1-L-bottles/year) | 27.33 ± 8.80 | 0–88 | 0 |
| Herbicides (1-L-bottles/year) | 25.73 ± 8.22 | 0–86 | 0 |
| Fungicides (1-L-bottles/year) | 21.47 ± 6.50 | 0–88 | 0 |
| Fertilizers (10-kg-bags/year) | 28.20 ± 8.89 | 0–104 | 0 |

### 3.3. Local Filters of Morphological Traits

Body length of males ($t = 3.47$, *d.f.* $= 13$, $p < 0.01$, $D^2 = 0.48$), and females ($t = 2.44$, *d.f.* $= 13$, $p = 0.02$, $D^2 = 0.32$) was positively explained by the use of fertilizers. Pronotum width of males ($t = 3.04$, *d.f.* $= 13$, $p < 0.01$, $D^2 = 0.42$), and females ($t = 3.19$, *d.f.* $= 13$, $p < 0.01$, $D^2 = 0.44$) was positively explained by the use of insecticides. Elytron length of both sexes was not significantly explained by any characteristic. Only rostrum length of females ($t = 4.66$, *d.f.* $= 13$, $p < 0.01$, $D^2 = 0.63$) was positively explained by palm density. Mesothorax depth of males ($t = 3.20$, *d.f.* $= 13$, $p < 0.01$, $D^2 = 0.44$), and females ($t = 2.24$, *d.f.* $= 13$, $p = 0.04$, $D^2 = 0.28$) was positively explained by palm diversity. Only the mesothorax depth of males was negatively explained by palm diversity abundance of pygmy date palm (*P. roebelenii*) ($t = -2.22$, *d.f.* $= 13$, $p = 0.04$, $D^2 = 0.28$).

## 4. Discussion

The south American palm weevil, *R. palmarum*, is a pest that threatens the phytosanitary quality, marketing, and profitability of diverse palm plantations in the Americas [37,41]. This study provides, for the first time, clear support for the hypothesis that local agricultural management of ornamental palm plantations filters variation in morphological traits of this pest populations. We can confirm that certain morphological traits vary widely among populations, and these are filtered by density and diversity of palms, and the use of insecticides and fertilizers. These findings may promote alternative strategies and optimize the barriers to limit populations of this devastating and uncontrolled pest [40], while maintaining productivity and profitability of ornamental palm plantations.

The variation range in morphological traits of *R. palmarum* has been reported in only two previous studies that considered a few adults of this sexually dimorphic species [43,44]. In this study, with a sample size of 4972 adults, we demonstrate that body length, pronotum width, rostrum length, and mesothorax depth vary widely among palm plantations. Variation ranges of these morphometric measurements are consistent with those reported in the taxonomic revision of this species, which were collected in diverse palm plantations in Bolivia, Costa Rica, El Salvador, and Paraguay [44]. A relatively recent report on mor-

phological variation and color polymorphism in only five adults collected in Colombia suggests that local environment plays a major ecological role in phenotypic plasticity of this pest [43]. In this sense, at the local scale, agricultural management characteristics of ornamental palm plantations may explain the high variation in morphological traits of *R. palmarum* considering the relatively small area (1 ha per plot) sampled in this study.

The quality of habitat resources, such as food, may select for phenotypic plasticity in insect growth and development regulation [24,39,62]. The present study shows that body length and mesothorax depth of adults of both sexes are greater in plantations with a high diversity of palm species and frequency of fertilizer use. Both variables modulate the nutritional environment in ornamental palm plantations since likely indicate availability, nutritional content, and predictability of host species [52,63,64]. This highlights the significance of palm species composition in plantations since certain palm species are potential or previously reported hosts for *R. palmarum* [31,65,66]. For this reason, a particular array of palm species within a plantation can favor or limit food quality of host palms [41], which make body length and mesothorax depth of weevils phenotypically plastic among populations. Since such morphometric measurements have been associated to foraging and feeding behavior and performance of several coleopteran species [46,51,52,67], diversification of palm species and intensification of fertilizer use may benefit the ability of *R. palmarum* populations to survive and reproduce in ornamental palm plantations.

The simultaneous cultivation in the same field of different crop species plays a strategic role when a polyphagous insect pest become pestiferous in newly invaded areas [8,68]. For instance, the most common crop species in the study area was, until 2009, the Mexican fan palm (*Washingtonia robusta* H. Wendl.), when *R. palmarum* was detected and its impact caused high economic losses to palm producers [41]. Subsequently, crop species' turnover increases among plantations in order to avoid further damage to ornamental palm crops [37]. Despite the dramatic reduction in the preferred host abundance for this pest, it prevails today with a progressive and excessive increase of damage. Our results show that rostrum length of females increases as palm density increases. Surprisingly, palm density was related to the abundance of queen palm (*S. romanzoffiana*), which represents 40 to 75% of the sampled palms in plantations. This rapid-growth species has been considered as pest-resistant by palm producers in the study area [41]. However, if *R. palmarum* possibly uses this common palm species as a novel host, then it may be filtering only those females with a longer rostrum. This morphological structure is related to the behavior of females, which digs holes in leaf petioles and in concealed locations of the palm crown to lay eggs [69]. Variations in this are due to selection pressure, assisted by the availability of food and nesting resources, and on reproductive traits [56,67]. In fact, increases in rostrum length indicate an effective use in the excavation and preparation of oviposition sites by females [55,56]. As rostrum length may reflect increases in the biotic potential and fertility of females, this morphological trait is crucial in the successful establishment of *R. palmarum* in ornamental palm plantations.

As found in other studies, agricultural intensification drives the degree to which traits of individuals or populations can rapidly adapt to new or changing environmental conditions [22,43,70]. In this study, ornamental palm plantations with high management intensity had a frequency of more than 50 insecticide bottles of 1-L capacity per year per plantation (Table 2). Deltamethrin (pyrethroid) and imidacloprid (neonicotinoid) are the most used and are only applied on healthy palm fronds using hand-held or backpack sprayers with long booms [41]. However, this practice is ineffective since adults do not remain for long periods of time on healthy palm fronds [37]. They are mainly attracted to diseased or wounded palms where they stop and reproduce [30,71]. Our analyses revealed that the pronotum width of both sexes is positively related to the use of insecticides. The internal muscles of the beetle pronotum is related to flight and movement [72,73], which enable them to adapt to different environmental conditions in diverse habitats [74]. Pronotum width has also been posited as a valuable trait in predicting population lifespan in some other beetle species [54]. Variation in pronotum width among populations may

be explained by the exposure to plantations contaminated with very toxic insecticides. In this sense, exposure time to synthetic insecticides is likely shorter in individuals with wider pronotum. In accordance with this, some studies have reported that this pest can prevail in agricultural areas with high intensity of insecticide use [75]. This is related to the prediction that agricultural intensification can create insect pests from herbivores, which illustrates that frequent insecticide use provides conditions permissive or conducive to the emergence and spread of pests [3,8]. This evidence unveils that intensifications of insecticide use promotes an environmental condition in ornamental palm plantations that filters adaptative morphological phenotypes of *R. palmarum* populations, resulting possibly in positive effects on lifespan populations [54].

Many studies have investigated the alternative strategies that farmers can use for controlling or suppressing this pest [46]. For instance, the tachinid flies *Billaea claripalpis* Wulp and *B. rhynchophorae* (Blanchard) (Diptera: Tachinidae) have been suggested as potential parasitoids [76,77]. There is also a report on larvae and adults of the rove beetle *Xanthopygus cognatus* Sharp (Coleoptera: Staphylinidae) predating eggs and larvae of this pest [78]. Surprisingly, the use of entomopathogenic fungi as control agents for this weevil has been little explored [76]. Recent studies present the isolation and identification of strains of *Metarhizium anisopliae* (Metchnikoff) Sorokin (Hypocreales: Clavicipitaceae), *Beauveria bassiana* (Bals.-Criv.) Vuill and *Trichoderma virens* (Mill.) from adults of *R. palmarum* and from the soil of coconut plantations (*Cocos nucifera* L.). These have been evaluated on larvae and adults in laboratory conditions and results have been highly effective [79]. The observed results for the use of synthetic agrochemicals suggest that policies and strategies must consider agroecological management practices to improve the efficacy and effectiveness of control of *R. palmarum* in ornamental palm plantations. There are some reports and field observations that in the study area, palm producers use some alternative biological (application of neem extracts), mechanical (adult capture by hand), and cultural (removal of diseased palms) practices to control *R. palmarum*. These strategies likely limit certain food, oviposition, and mating resources that prevent this pest from population outbreaks [46]. Therefore, using alternative management practices might be a viable and safe solution to directly impact and decrease the abundance of this weevil. It might also be effective for promoting agroecological transition in ornamental palm plantations.

## 5. Conclusions

This is the first ecological study of the south American palm weevil (*Rhynchophorus palmarum*) that provides new insights into the current intensive management of ornamental palm plantations. Far from controlling, benefits this economically important pest [80,81]. Given the current geographic expansion, demographic outbreak, and economic impact of *R. palmarum* [46], it is worth noting how local conditions of plantations filter integrated, adaptative, and environment-specific phenotypes [82]. Particularly, biophysical structure of plantations and agrochemical use should be addressed to explain morphological plasticity observed in this pest [43,44,66]. Palm density and diversity and the use of fertilizers and insecticides represent resource availability (food, oviposition, or mating) and environmental conditions that drive or constraint morphological traits, and possible another population attributes of *R. palmarum*.

**Supplementary Materials:** The following supporting information can be downloaded at: https://www.mdpi.com/article/10.3390/agronomy12102371/s1, Table S1: Species composition and mean density (individuals/ha) of planted ornamental palms in the studied permanent sampling plots where *Rhynchophorus palmarum* adults were captured in central Veracruz, Mexico, Table S2: Farmer semi-structured survey to identify the intensity of agricultural management practices for controlling pests, weeds, fungi, and enrich soil fertility in ornamental palm crops in central Veracruz, Mexico.

**Author Contributions:** Conceptualization, M.P.-M., C.L.-H., M.A.G.-M., J.M.-G., R.L.-C., E.P.-P. and R.S.-L.; methodology, M.P.-M., C.L.-H. and M.A.G.-M.; formal analysis, E.P.-P. and M.A.G.-M.; investigation, M.P.-M. and M.A.G.-M.; resources, J.M.-G. and M.A.G.-M.; data curation, M.A.G.-M.; writing—original draft preparation, M.P.-M. and M.A.G.-M.; writing—review and editing, M.A.G.-M.; visualization, M.A.G.-M., J.M.-G., R.L.-C., E.P.-P. and R.S.-L.; supervision, M.A.G.-M.; project administration, M.A.G.-M.; funding acquisition, J.M.-G. and M.A.G.-M. All authors have read and agreed to the published version of the manuscript.

**Funding:** This research was supported by grants from the Direccion General de Investigaciones of Universidad Veracruzana (Desarrollo de la Investigacion program, Beca Apoyo al Sistema Nacional de Investigadores scholarship, 47041). M.P.-M. received a doctoral fellowship (2019-000037-02NACF-27629/863446/757058, Convocatoria 004173) from Consejo Nacional de Ciencia y Tecnologia (CONACyT) during the duration of this study.

**Institutional Review Board Statement:** Not applicable.

**Informed Consent Statement:** Not applicable.

**Data Availability Statement:** Morphometric traits data of adult weevils presented in this study are available on request from the corresponding author.

**Acknowledgments:** We are grateful for the Sara Paulet Hernandez Silvestre and Juan Jose Soto de Aquino for technical assistance in the field and the processing of samples. We thank the anonymous reviewers for their valuable comments and suggestions to improve the manuscript.

**Conflicts of Interest:** The authors declare no conflict of interest.

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
