# Peer review of "Local Agricultural Management Filters Morphological Traits of the South American Palm Weevil (Rhynchophorus palmarum L.; Coleoptera: Curculionidae) in Ornamental Palm Plantations"

_agronomy, doi:10.3390/agronomy12102371_

Round 1

Reviewer 1 Report

The palm weevil Rhynchophorus palmarum is an important pest of palm trees in tropical and subtropical America, that has been devoted to several scientic studies. This contribution has addressed a new point of interest, that is the phenotypic plasticity of the species.

The variation in several morphological traits is probably related to some characteristics of importance for pest management, being the main strength  of this study.

The authors have studied a good samples of specimens to draw preliminary conclusions on the relation specific traits/ local characteristics of agricultural management. Their experimental designs and analyses are correct, and useful as a model for studies on phenotypic plasticity in other weevil pests. 

Author Response

Dear Reviewer #1,

Enclosed please find our manuscript agronomy-1880686 entitled “Local Agricultural Management Filters Morphological Traits of the South American Palm Weevil (Rhynchophorus palmarum L.; Coleoptera: Curculionidae) in Ornamental Palm Plantations” by Moisés Ponce-Méndez, Miguel A. García-Martínez, Joaquín Murguía-González, Ricardo Serna-Lagunes, Rodrigo Lasa-Covarrubias, Ehdibaldo Presa-Parra y Carlos Llarena-Hernández. We are re-submitting for publication in Agronomy following major revisions.

The current version of the manuscript takes into consideration all the suggestion offered by you, which have greatly improved the clarity and flow of the paper. As Editor indicated, we revised our manuscript using a word processing program, and we highlighted the changes within the document by using the track changes mode in MS Word. Our specific responses to the comments and suggestions are provided at following.

Please let me know if there are any other questions or concerns regarding this manuscript. My co-authors and I look forward to hearing from you soon.

Sincerely, Miguel A. García-Martínez

Comments and Suggestions

Response

The palm weevil Rhynchophorus palmarum is an important pest of palm trees in tropical and subtropical America, that has been devoted to several scientific studies. This contribution has addressed a new point of interest, that is the phenotypic plasticity of the species.

We are grateful for the comment. In fact, we hope that this study improves the knowledge of this pestiferous weevil.

The variation in several morphological traits is probably related to some characteristics of importance for pest management, being the main strength of this study.

We appreciate this comment. In this study we present an inference of the relation of morphological traits and management characteristic to provide information for pest management plans in the study area and other ones.

The authors have studied a good sample of specimens to draw preliminary conclusions on the relation specific traits/ local characteristics of agricultural management.

We are grateful for your appreciation. Yes, we study 15 permanent plot along an entire sampling year and also we measured, by taking pictures, 5 morphological traits in all collected weevils.

Their experimental designs and analyses are correct, and useful as a model for studies on phenotypic plasticity in other weevil pests.

We appreciate these comments. We hope that this study may serve a base line in the development of ecological studies in R. palmarum, another Rhynchophorus species and another pest weevils.

Reviewer 2 Report

Overall, this manuscript is interesting and addresses a novel aspect for an economic important insect pest on ornamental palms in Mexico. 

This manuscript needs some small paragraphs in introduction and discussion that can improve its quality (see review report). 

In addition, address alternative strategies of control different to the use of insecticides that farmers can use for suppress this pest, while reduce the extremely high number of insecticide sprays per year. 

Author Response

Dear Reviewer #2,

Enclosed please find our manuscript agronomy-1880686 entitled “Local Agricultural Management Filters Morphological Traits of the South American Palm Weevil (Rhynchophorus palmarum L.; Coleoptera: Curculionidae) in Ornamental Palm Plantations” by Moisés Ponce-Méndez, Miguel A. García-Martínez, Joaquín Murguía-González, Ricardo Serna-Lagunes, Rodrigo Lasa-Covarrubias, Ehdibaldo Presa-Parra y Carlos Llarena-Hernández. We are re-submitting for publication in Agronomy following major revisions.

The current version of the manuscript takes into consideration all the suggestion offered by you, which have greatly improved the clarity and flow of the paper. As Editor indicated, we revised our manuscript using a word processing program, and we highlighted the changes within the document by using the track changes mode in MS Word. Our specific responses to the comments and suggestions are provided at following.

Please let me know if there are any other questions or concerns regarding this manuscript. My co-authors and I look forward to hearing from you soon.

Sincerely, Miguel A. García-Martínez

Comments and Suggestions

Response

Overall, this manuscript is interesting and addresses a novel aspect for an economic important insect pest on ornamental palms in Mexico.

We are grateful for your appreciation.

This manuscript needs some small paragraphs in introduction and discussion that can improve its quality (see review report).

We appreciate this suggestion, and we made all suggested changes indicated in review report.

In addition, address alternative strategies of control different to the use of insecticides that farmers can use for suppressing this pest, while reduce the extremely high number of insecticide sprays per year.

Done. We added a paragraph at the end of discussion to provide readers these alternatives:

Many studies have investigated the alternative strategies that farmers can use for controlling or suppressing this pest [46]. For instance, the tachinid flies Billaea claripalpis Wulp and B. rhynchophorae (Blanchard) (Diptera: Tachinidae) have been suggested as potential parasitoids [77,78]. There is also a report on larvae and adults of the rove beetle Xanthopygus cognatus Sharp (Coleoptera: Staphylinidae) predating eggs and larvae of this pest [79]. Surprisingly, the use of entomopathogenic fungi as control agents for this weevil has been little explored [77]. Recent studies present the isolation and identification of strains of Metarhizium anisopliae (Metchnikoff) Sorokin (Hypocreales: Clavicipitaceae), Beauveria bassiana (Bals.-Criv.) Vuill and Trichoderma virens (Mill.) from adults of R. pal-marum and from the soil of coconut plantations (Cocos nucifera L.). These have been evaluated on larvae and adults in laboratory conditions and results have been highly effective [80]. The observed results for the use of synthetic agrochemicals suggest that policies and strategies must consider agroecological management practices to improve the efficacy and effectiveness of control of R. palmarum in ornamental palm plantations. There are some reports and field observations that in the study area, palm producers use some alternative biological (application of neem extracts), mechanical (adult capture by hand), and cultural (removal of diseased palms) practices to control R. palmarum. These strategies likely limit certain food, oviposition, and mating resources that prevent this pest from population out-breaks [46]. Therefore, using alternative management practices might be a viable and safe solution to directly impact and decrease abundance of this weevil, it also might be effective for promoting agroecological transition in ornamental palm planta-tions.

Line 32. Introduction. Two important aspects need to be addressed in introduction: First, since your topic is related to ornamental palm plantations in Mexico, it is relevant to include a small paragraph about the socio-economic importance of this crop. For example: area planted on ornamental palms, value of the marketing (palms sold from nurseries etc.), generation of employ, number of producers, etc.

We thank to the reviewer for this suggestion to improve our manuscript. We added this information to the manuscript:

The commercial exploitation of ornamental palms plays a valuable role in the economy of international trade [35]. Plantations represent a significant source of income for small farmers due to palm health, quality, and high aesthetic value [36]. This crop is rare throughout Mexico since some producers are entering the market. Their current planted area is only represented in Baja California Sur (82 ha), Chiapas (16 ha), San Luis Potosí (734 ha), and Veracruz (711 ha) with ca. 468 cultivated palm species. Particularly, in Veracruz there are approximately 40 cultivated palm species distributed in 19 municipalities, where there are small farms from 0.56 to 3.8 ha. Total area of this crop is 1,559 ha be-longing to 2,768 farms. Produced ornamental palms are commercialized to costal touristic areas in Mexico [36]. However, ornamental palm agribusinesses, and particularly of central Veracruz, have reported yield losses of approximately $6,000.00 USD/ha caused by R. palmarum [37]. Although management of this pest has intensified using broad-spectrum insecticides mixed with synthetic baits and/or food attractants [37–39], populations are currently expanding their geographical range and exploiting novel palm hosts [40–42]. In fact, density of certain ornamental palm species may modulate abun-dance population of this pest in space and time [41]. Given this context, agrochemical use and crop density and diversity in ornamental palm plantations might improve R. pal-marum populations, likely their morphological traits [43], which promotes their survival, spread and establishment in vast agricultural areas.

Since you are talking about Rhynchophorus palmarum, its relevant also address basic aspects about its biology (life cycle), ecology and behaviors plus a short description of damages caused on palms. What is the economic loses estimation in Mexico or in Veracruz caused by this pest? Only two short paragraphs addressing those two relevant aspects

We thank to the reviewer for this suggestion to improve our manuscript. We added this information to the manuscript:

The south American palm weevil, Rhynchophorus palmarum (L.) (Coleoptera: Cur-culionidae), is an invasive and, polyphagous pest in the Americas [26–30]. Adults have been reported as phytophagous of economic important crops such as avocado (Persea americana Mill.; Laurales: Lauraceae), banana (Musa × paradisiaca L.; Zingiberales: Mu-saceae), cocoa (Theobroma cacao L.; Malvaceae: Theobromeae) and sugar cane (Saccharum officinarum L.; Poales: Poaceae) [31,32]. However, it has been highlighted as the primary pest of palms (Arecales: Arecaceae) cultivated for food or ornamental purposes [32]. The larval phase of this weevil directly damages palms because they feed on meristematic tissue and can tunnel palm stems. This prevents palms from producing new fronds, destroys the growth point, damages the lower stem and rhizomes, and eventually causes palm death [28,33]. Adults also indirectly damage since they act as vectors of the causal agents of the bud rot disease (Phytophthora palmivora Butler; Peronosporales: Pythiaceae), and the red ring disease (Bursaphelenchus cocophilus (Cobb); Tylenchida: Aphelenchoi-didae) [29,30,34]. Given this biological, ecological, and phytosanitary importance, R. palmarum represents a devastation pest in several palm plantations, mainly ornamental ones.

Line 67 a disease vector of ornamental palms. What specific disease is transferred by this insect pest on palms?

We stated in manuscript following information:

The south American palm weevil, Rhynchophorus palmarum (L.) (Coleoptera: Cur-culionidae), is an invasive and, polyphagous pest in the Americas [26–30]. Adults have been reported as phytophagous of economic important crops such as avocado (Persea americana Mill.; Laurales: Lauraceae), banana (Musa × paradisiaca L.; Zingiberales: Mu-saceae), cocoa (Theobroma cacao L.; Malvaceae: Theobromeae) and sugar cane (Saccharum officinarum L.; Poales: Poaceae) [31,32]. However, it has been highlighted as the primary pest of palms (Arecales: Arecaceae) cultivated for food or ornamental purposes [32]. The larval phase of this weevil directly damages palms because they feed on meristematic tissue and can tunnel palm stems. This prevents palms from producing new fronds, destroys the growth point, damages the lower stem and rhizomes, and eventually causes palm death [28,33]. Adults also indirectly damage since they act as vectors of the causal agents of the bud rot disease (Phytophthora palmivora Butler; Peronosporales: Pythiaceae), and the red ring disease (Bursaphelenchus cocophilus (Cobb); Tylenchida: Aphelenchoi-didae) [29,30,34]. Given this biological, ecological, and phytosanitary importance, R. palmarum represents a devastation pest in several palm plantations, mainly ornamental ones.

Line 110, 119, 123. Ornamental palm plantations. In the methodology you are talking about ornamental palm plantations, which is correct, however, it was not very clear what is the socioeconomic importance of this crop in Mexico. For this reason, is relevant to address this aspect in the introduction.

We appreciate this observation. In the manuscript we indicated that:

The commercial exploitation of ornamental palms plays a valuable role in the economy of international trade [35]. Plantations represent a significant source of income for small farmers due to palm health, quality, and high aesthetic value [36]. This crop is rare throughout Mexico since some producers are entering the market. Their current planted area is only represented in Baja California Sur (82 ha), Chiapas (16 ha), San Luis Potosí (734 ha), and Veracruz (711 ha) with ca. 468 cultivated palm species. Particularly, in Veracruz there are approximately 40 cultivated palm species distributed in 19 municipalities, where there are small farms from 0.56 to 3.8 ha. Total area of this crop is 1,559 ha be-longing to 2,768 farms. Produced ornamental palms are commercialized to costal touristic areas in Mexico [36]. However, ornamental palm agribusinesses, and particularly of central Veracruz, have reported yield losses of approximately $6,000.00 USD/ha caused by R. palmarum [37]. Although management of this pest has intensified using broad-spectrum insecticides mixed with synthetic baits and/or food attractants [37–39], populations are currently expanding their geographical range and exploiting novel palm hosts [40–42]. In fact, density of certain ornamental palm species may modulate abun-dance population of this pest in space and time [41]. Given this context, agrochemical use and crop density and diversity in ornamental palm plantations might improve R. pal-marum populations, likely their morphological traits [43], which promotes their survival, spread and establishment in vast agricultural areas.

Line 141. A semi -structured interview with farmer managers etc. (Table S2). I think, your questionnaire was correct on questions related with the use of insecticides, fungicides, herbicides and fertilizers. However, for the response to those questions, I am not sure if those responses reflect the real situation of the agronomic management given by farmers. I do not know it you did additional evaluations for verification of farmers response. For example, in the response about use of fertilizers (0 – 104) average 28.20 per year (table 2), I think this is an exaggeration, similar situation related with the use of insecticides, herbicides and fungicides. However, I do not know how is the ornamental palm crop industry in Mexico. For this reason, is important to include additional sources of information for corroboration.

Is there any additional information form this location (Extension Service Office), or from the Department of Agriculture, that reflect your finding or that opposed them?

We apologize to the reviewer for not clarifying the information provided. We conducted an interview to find out the number of polyethylene bags with a capacity of 10 kg fertilizer (N, P, K) applied during the sampling year in each of the plantations. Then the producers who were interviewed told us the number of bags of fertilizer used and these were corroborated by checking the bags discarded in each plantation. In this sense, we did not sample the number of times that fertilizations were carried out in the plantation during a year, but rather the quantity or the number of bags applied. Given this context, plantations with high intensity management have a total 104 10-kg-bags of fertilizer per year or 1.14 ton/year. The same happened with the use of insecticides, herbicides, and fungicides. In this case, we interview the producers to find out the number of 1-L-bottles that they apply throughout the year. We try to corroborate the information provided by the producers and several times we were able to observe the bottles of such agrochemicals abandoned in the center of the plantation. Given this context, plantations with high intensity management showed a total of 88 1-L-bottles/year or 7.3 1-L-bottles/month or 1.8 1-L-bottles/week. For this reason, we apologize and try to correct that information in the manuscript for clarity. Without you important suggestions it could not be possibly.

Line 244. Table 2. In this table you need to include the Mode (number that appears the most often for the set) for the use of insecticides, fungicides, herbicides, and fertilizers. Keep the mean and the range. In addition, for the mean in all variables you need to include the SEM (Standard Error of the Mean). This is just descriptive statist.

We thank to the reviewer for this suggestion to improve our manuscript. We added this information to the Table 2 in a column with the header Mode and next to the mean we added ± SEM.

Line 147. Trap-food-attractant. Take a photograph about this specific trap and include it in the manuscript is recommended.

We have made the suggested changes. Now Fig. 2 is a figure composed of two pictures where there is a frontal view of the trap set in field and the revised trap after 4 weeks of exposure.

Line 161. Each trap. What was the number of traps installed in each plot and for the full survey?

We are grateful for this comment, we stated in the manuscript that:

In each plot were set 5 traps, considering 15 sampling plots a total of 75 traps were set each month, resulting in a total of 900 traps for the entire sampling.

In addition, how many samplings (evaluations) were conducted during the full field survey.

We stated in manuscript that:

In each plot were set 5 traps, considering 15 sampling plots a total of 75 traps were set each month, resulting in a total of 900 traps for the entire sampling of 12 months.

Line 215, 216. Include both the total number of males and females captured and their corresponding percentage.

We thank to the reviewer for this suggestion to improve our manuscript. We added this information to the manuscript:

A total of 3,077 (61.8%) of the captured weevils were male, and 1,895 (38.1%) were female.

Line 230. Table 1. Include (N), the number of males, females and both sexes.

Done. We added to the Table 1 a row where is indicated the sampling size as abundance measured as individuals. In fact, this N represents the number of measured weevils.

Lines 239-242. Those results are extremely high. However, if those are the data keep them, but in discussion include additional information (references) that support your findings or that opposed them.

We apologize to the reviewer for not clarifying the information provided. We conducted an interview to find out the number of polyethylene bags with a capacity of 10 kg fertilizer (N, P, K) applied during the sampling year in each of the plantations. Then the producers who were interviewed told us the number of bags of fertilizer used and these were corroborated by checking the bags discarded in each plantation. In this sense, we did not sample the number of times that fertilizations were carried out in the plantation during a year, but rather the quantity or the number of bags applied. The same happened with the use of insecticides, herbicides, and fungicides. In this case, we interview the producers to find out the number of 1l bottles that they apply throughout the year. We try to corroborate the information provided by the producers and several times we were able to observe the bottles of such agrochemicals abandoned in the center of the plantation. For this reason, we apologize and try to correct that information in the manuscript for clarity.

Line 248 -255. For all those variables if correlations were conducted could be important to include the R2 value (correlation coefficient). It is important to see the strength for each relationship.

We thank to the reviewer for this suggestion to improve our manuscript. In fact, we performed regression models fitting a predictor (local management characteristic) with a response variable (morphological trait) using generalized linear models. The goodness-of-fit of the models was estimated as the explained deviance for each model using the modEvA package for R version 3.2.2 [58].

We added this information to the manuscript. The goodness-of-fit of each significant model is indicated in parenthesis as the proportion of deviance (D2) explained by each model.

Reviewer 3 Report

This is a very interesting study about the effect of agricultural management on the phenotypic plasticity of a weevil pest species. It is well written, but I found a few aspects to consider in a revised version:

In lines 139, 140 it is stated (citing the work of Landero-Torres et al. 2015) that Syagrus romanzoffiana (and Phoenix have not been reported as hosts of R. palmarum, but I´ve seen several reports (e.g., https://gd.eppo.int/taxon/RHYCPA/hostshttp://inta.gob.ar/sites/default/files/inta-hd_46-_el_picudo_rhynchophorus_palmarum_mata_palmeras_en_corrientes.pdf) that inform about host-plant association and damages to Syagrus romanzoffiana by R. palmarum. Also, although Landero-Torres et al. (2015) documented that the palm is more resistant, it is understood that it does host R. palmarum. Please check the available literature accurately to verify this and reword the phrase if needed.

Fig.2 (e). It must indicate the morphological trait: Mesothorax depth. However, the yellow line is clearly on the Prothorax, so it needs to be corrected (moved to the right, to indicate the mesothorax, as explained in lines 179, 180).

Author Response

Dear Reviewer #3, 

Enclosed please find our manuscript agronomy-1880686 entitled “Local Agricultural Management Filters Morphological Traits of the South American Palm Weevil (Rhynchophorus palmarum L.; Coleoptera: Curculionidae) in Ornamental Palm Plantations” by Moisés Ponce-Méndez, Miguel A. García-Martínez, Joaquín Murguía-González, Ricardo Serna-Lagunes, Rodrigo Lasa-Covarrubias, Ehdibaldo Presa-Parra y Carlos Llarena-Hernández. We are re-submitting for publication in Agronomy following major revisions.

The current version of the manuscript takes into consideration all the suggestion offered by you, which have greatly improved the clarity and flow of the paper. As Editor indicated, we revised our manuscript using a word processing program, and we highlighted the changes within the document by using the track changes mode in MS Word. Our specific responses to the comments and suggestions are provided at following.

Please let me know if there are any other questions or concerns regarding this manuscript. My co-authors and I look forward to hearing from you soon.

Sincerely, Miguel A. García-Martínez

Comments and Suggestions

Response

This is a very interesting study about the effect of agricultural management on the phenotypic plasticity of a weevil pest species. It is well written, but I found a few aspects to consider in a revised version:

We appreciate this comment. We conducted all suggestions provided by you.

In lines 139, 140 it is stated (citing the work of Landero-Torres et al. 2015) that Syagrus romanzoffiana (and Phoenix have not been reported as hosts of R. palmarum, but I´ve seen several reports (e.g., https://gd.eppo.int/taxon/RHYCPA/hosts; http://inta.gob.ar/sites/default/files/inta-hd_46_el_picudo_rhynchophorus_palmarum_mata_palmeras_en_corrientes.pdf) that inform about host-plant association and damages to Syagrus romanzoffiana by R. palmarum. Also, although Landero-Torres et al. (2015) documented that the palm is more resistant, it is understood that it does host R. palmarum. Please check the available literature accurately to verify this and reword the phrase if needed.

We thank to the reviewer for this suggestion to improve our manuscript. We added this information to the manuscript and reworded the sentence.

Fig.2 (e). It must indicate the morphological trait: Mesothorax depth. However, the yellow line is clearly on the Prothorax, so it needs to be corrected (moved to the right, to indicate the mesothorax, as explained in lines 179, 180).

We have made the suggested changes to Fig. 2. This new figure indicates mesothorax depth as a yellow line between the lower and the upper surface at the base level of elytra in profile view.  

Round 2

Reviewer 2 Report

I have reviewed this manuscript for second time.  I see a lot of improvements addressed by authors.  Only very small corrections are needed:

1- In Abstract include  Veracruz, Mexico  (Line 19.) 

2- In Discussion, (Line 374).  You talk about frequency of insecticide  applications  (> 50 applications per year), (Table 2).  Correction:  L-bottle per year per farm. 

I have no more comments or corrections.

Good job! 

Author Response

Dear Reviewer,

We are very grateful for your revisions that improves highly the flow and clarity of our manuscript. We performed all suggestions that you provided.

Suggestion

Response

1- In Abstract include  Veracruz, Mexico (Line 19.)

Done. Now in abstract there is the following sentence:

This study assesses the influence of biophysical structure of ornamental palm plantations and agrochemical use on morphological traits of adults in 15 permanent plots of ornamental palm plantations in Veracruz, Mexico.

2- In Discussion, (Line 374).  You talk about frequency of insecticide applications (> 50 applications per year), (Table 2).  Correction:  L-bottle per year per farm.

Done. Now in abstract there is the following sentence:

In this study, ornamental palm plantations with high management intensity had a frequency of more than 50 insecticide bottles of 1-L capacity per year per plantation.